# Attenuated Total Reflection Fourier-Transform Infrared Spectral Discrimination in Human Tissue of Oesophageal Transformation to Adenocarcinoma

**DOI:** 10.3390/jpm13081277

**Published:** 2023-08-20

**Authors:** Ishaan Maitra, Camilo L. M. Morais, Kássio M. G. Lima, Katherine M. Ashton, Danielle Bury, Ravindra S. Date, Francis L. Martin

**Affiliations:** 1School of Pharmacy and Biomedical Sciences, University of Central Lancashire, Preston PR1 2HE, UK; 2Institute of Chemistry, Biological Chemistry and Chemometrics, Federal University of Rio Grande do Norte, Natal 59078-970, Brazil; camilomorais1@gmail.com (C.L.M.M.); kassiolima@gmail.com (K.M.G.L.); 3Center for Education, Science and Technology of the Inhamuns Region, State University of Ceará, Tauá 63660-000, Brazil; 4Lancashire Teaching Hospitals NHS Foundation Trust, Royal Preston Hospital, Preston PR2 9HT, UK; katherine.ashton@lthtr.nhs.uk (K.M.A.); ravidate@hotmail.com (R.S.D.); 5Department of Cellular Pathology, Blackpool Teaching Hospitals NHS Foundation Trust, Blackpool FY3 8NR, UK; danielle.bury@nhs.net

**Keywords:** oesophageal cancer, ATR-FTIR, tissue, multivariate classification

## Abstract

This study presents ATR-FTIR (attenuated total reflectance Fourier-transform infrared) spectral analysis of ex vivo oesophageal tissue including all classifications to oesophageal adenocarcinoma (OAC). The article adds further validation to previous human tissue studies identifying the potential for ATR-FTIR spectroscopy in differentiating among all classes of oesophageal transformation to OAC. *Tissue spectral analysis* used principal component analysis quadratic discriminant analysis (PCA-QDA), successive projection algorithm quadratic discriminant analysis (SPA-QDA), and genetic algorithm quadratic discriminant analysis (GA-QDA) algorithms for variable selection and classification. The variables selected by SPA-QDA and GA-QDA discriminated tissue samples from Barrett’s oesophagus (BO) to OAC with 100% accuracy on the basis of unique spectral “fingerprints” of their biochemical composition. Accuracy test results including sensitivity and specificity were determined. The best results were obtained with PCA-QDA, where tissues ranging from normal to OAC were correctly classified with 90.9% overall accuracy (71.4–100% sensitivity and 89.5–100% specificity), including the discrimination between normal and inflammatory tissue, which failed in SPA-QDA and GA-QDA. All the models revealed excellent results for distinguishing among BO, low-grade dysplasia (LGD), high-grade dysplasia (HGD), and OAC tissues (100% sensitivities and specificities). This study highlights the need for further work identifying potential biochemical markers using ATR-FTIR in tissue that could be utilised as an adjunct to histopathological diagnosis for early detection of neoplastic changes in susceptible epithelium.

## 1. Introduction

Vibrational spectroscopic analysis of human tissue combined with chemometric methods including principal component analysis (PCA) and successive projection algorithm (SPA) has been utilised to identify discriminate spectra between benign and malignant tissue [1]. Over the past 10 to 15 years, this technology has identified significant differences in diagnostic bands between the spectra of malignant and corresponding normal tissues [2,3].

In the gastrointestinal (GI) tract, vibrational spectroscopy techniques have been able to discriminate between benign and neoplastic pathology in stomach [4] and intestinal tissue [5,6]. Spectroscopic analysis in oesophageal tissue is gaining momentum in identifying subtle spectral differences between benign and malignant disease. Although using vibrational spectroscopy in analysing oesophageal tissue is not new [7,8,9,10], it has great potential moving forwards.

With oesophageal adenocarcinoma (OAC) pathogenesis, the role of dysplastic processes in Barrett’s oesophagus (BO) is pertinent. Barrett’s oesophagus (BO) results in chronic inflammation from gastro-oesophageal reflux disease (GORD) and is the only known precursor to OAC [11]. Establishing dysplasia in Barrett’s oesophagus at an earlier stage would enable less invasive treatment options being utilised compared to the high morbidity and mortality associated with OAC and its subsequent treatment [12].

The gold standard in diagnostics in oesophageal disease is white-light oesophagogastroduodenoscopy (OGD) and tissue biopsy. In addition to being an invasive process, other disadvantages include variability in biopsy harvest, as well as endoscopist sampling errors [13,14]. In the context of malignant disease, the only available, reliable marker which can sub-select progression to OAC is low-grade dysplasia (LGD) [7]. There is evidence of significant intra- and inter-histopathologist bias in tissue specimen analysis, resulting in inaccurate diagnoses of dysplasia [13]. ATR-FTIR spectral analysis of ex vivo tissue could be utilised as an additional tool to aid histopathologists in confirming dysplasia earlier, allowing less invasive treatment options to be utilised at this point.

Biomedical point-of-care applications have not been introduced into clinical settings despite significant FTIR-based improvements. Issues remain related to batch processing and standardising tissue sections for comparable, reproducible analysis [15]. Lastly, large-volume spectral dataset analysis is laborious and time-consuming. These processes would need significant improvement before clinical implementation.

This paper presents an ex vivo study of all classifications to OAC including outlining tissue identification, spectral analysis. The study adds further validation to previous human studies identifying the potential for ATR-FTIR spectroscopy in differentiating among all classes in oesophageal transformation to OAC (normal; inflammatory; Barrett’s; low-grade dysplasia, LGD; high-grade dysplasia, HGD; OAC) and the use of this technology as an adjunct to aid histopathologists in diagnosing dysplasia accurately. Methods of tissue acquisition, sample preparation, and tissue analysis using chemometric methods are described.

## 2. Material and Methods

### 2.1. Sample Collection

Ethical approval was granted from both the parent University (STEMH 909 application) and by the East of England—Cambridge Central Research Ethics Committee from 2015 (archival gastro-intestinal tissue, blood, saliva, and urine collection; REC reference: 18/EE/0069; IRAS project ID: 242639). Patients were identified prospectively from pathology in-house hospital databases, and consent for tissue was taken between October 2017 and June 2019 in a clinic or an endoscopy setting. Paraffin-embedded samples were coded as normal squamous epithelium, squamous epithelium with inflammation, intestinal metaplasia, low-grade dysplasia (LGD), high grade dysplasia (HGD), and oesophageal adenocarcinoma (OAC) according to local departmental protocols. Paraffin wax embedding ensures durability for long-term storage without biomolecular tissue deterioration [16,17]. Paraffin embedding affects bands at 1465 cm^−1^ (associated with aromatic structure) [18] in the fingerprint region and bands at 2850, 2918, and 2956 cm^−1^ [19]. De-parrafinisation was, hence, performed after cutting prior to commencing ATR-FTIR measurements using local hospital protocols using xylene and ethanol as per institution protocol. Nevertheless, the deparaffination procedure causes alterations in the samples such as paraffin residuals [20]; thus, the models constructed with these samples can only be tested with deparaffined samples, not fresh or frozen tissues.

Contiguous sections with 4 µm thickness measuring 5 mm × 5 mm were prepared on FisherBrand™ slides and utilised so that each section closely resembled other sections, thus ensuring accurate correlation. A separate consultant histopathologist identified sections of the cut biopsies for an overall accurate representative analytical study of the tissue. This was to ensure that spectral measurements would be taken from the appropriate area. The specimens were compared with H&E (haematoxylin and eosin) specimens and specific random areas in this field chosen for point spectral analysis. British Society of Gastroenterology guidelines [21] state that all cases of suspected dysplasia are to be reviewed by a second GI pathologist. All slides were left to desiccate prior to transportation to the spectroscopy laboratory for analysis. Tissue samples were stored in a de-humidified container to prevent condensation and physical damage [22].

### 2.2. ATR-FTIR Spectroscopy

Sample interrogation was performed at the Biomedical Research Laboratory at the University of Central Lancashire (UK). Histological diagnoses were unknown to those who performed spectroscopy. Spectra were obtained using a Bruker TENSOR 27 FTIR spectrometer with a Helios ATR attachment containing a diamond crystal (Bruker Optics Ltd., Coventry, UK) and operated using OPUS 6.5 software. Spectra were acquired from 10 independent random sample locations pre-defined after comparing with H. Data acquisition parameters were as follows: 8 cm^−1^ spectral resolution, 32 scans, 6 mm aperture setting, and 2× zero-filling factors. The diamond crystal was washed with distilled water and dried with tissue paper between each sample and before each new slide. A background absorption spectrum was acquired prior to each new sample for atmospheric correction [22] (Figure 1).

### 2.3. Data Analysis and Chemometric Methods

The data import, pre-treatment, and construction of chemometric classification models were implemented in MATLAB R2014a software (MathWorks, Natick, MA, USA) using the PLS Toolbox version 7.9.3 (Eigenvector Research, Inc., Washington, DC, USA) and custom-made routines. Raw spectra were pre-processed by cutting between 1800 and 900 cm^−1^ (235 wavenumbers), followed by rubber-band baseline correction and normalisation to the amide I peak (1700 to 1600 cm^−1^) [23]. Monte Carlo cross-validation [24] was employed to validate the chemometric models. In this validation process, some of the samples were left out for validation in an exhaustive iterative process with several permutations. In our case, 20% of samples were left out for validation during 1000 iterations. The multivariate classification models used were principal component analysis quadratic discriminant analysis—PCA-QDA, successive projections algorithm quadratic discriminant analysis—SPA-QDA, and genetic algorithm quadratic discriminant analysis—GA-QDA. The optimum number of variables for SPA-QDA and GA-QDA was determined according to an average risk G of misclassification. Such a cost function was calculated with a random validation set as follows:(1)$$G=\frac{1}{N_{V}}\sum_{n=1}^{N_{V}}g_{n}$$where $N_{V}$ is the number of validation spectra, and $g_{n}$ is defined as
(2)$$g_{n}=\frac{r^{2}\left(x_{n},m_{I(n)}\right)}{\min_{I(m)\neq I(n)}r^{2}\left(x_{n},m_{I(m)}\right)}$$

In the above definition, the numerator is the squared Mahalanobis distance between object $x_{n}$ (of class index I(n)) and the sample mean $m_{I(n)}$ of its true class. The denominator in Equation (2) corresponds to the squared Mahalanobis distance between object $x_{n}$ and the centre of the closest wrong class, $m_{I(m)}$. The minimum value of the cost function (maximum fitness) is achieved when the selected variables from the original data are as close as possible to the true class and as distant as possible from the wrong class according to the validation samples. The GA routine was carried out with 100 generations with 200 chromosomes each. Crossover and mutation probabilities were set to 60% and 1%, respectively. The algorithm was subsequently repeated three times from different random initial populations. The best solution (in terms of the fitness value) resulting from the three realisations of the GA was employed. 

The QDA classification score ($Q_{ik}$) was estimated using the variance–covariance matrix for each class *k* and an additional natural logarithm term, as follows:(3)$$Q_{ik}={\left(x_{i}-{\bar{x}}_{k}\right)}^{T}\Sigma_{k}^{-1}\left(x_{i}-{\bar{x}}_{k}\right)+\log_{e}\left|\Sigma_{k}\right|-2\log_{e}\pi_{k}$$where $x_{i}$ is a vector with the input variables for sample *i*, ${\bar{x}}_{k}$ is the mean vector of class *k*, $\Sigma_{k}$ is the variance–covariance matrix of class *k*, $\log_{e}\left|\Sigma_{k}\right|$ is the natural logarithm of the determinant of the variance–covariance matrix of class *k*, and $\log_{e}\pi_{k}$ is the natural logarithm of the prior probability term of class *k*. QDA forms a separated variance model for each class and does not assume that different classes have similar variance–covariance matrices. This is different to what is assumed by linear discriminant analysis (LDA) [25].

The calculation of figures of merit is a recommended standard practice to test model performance [26,27]. Furthermore, accuracy, sensitivity (portion of positive samples correctly classified), specificity (portion of negative samples correctly classified), and the F-score (a general measurement of the model accuracy) were calculated using the following equations:(4)$$\mathrm{Sensitivity}\;\left(\%\right)=\frac{TP}{TP+FN}\times 100$$
(5)$$\mathrm{Specificity}\;\left(\%\right)=\frac{TN}{TN+FP}\times 100$$
(6)$$F\text{-}\mathrm{score}=\frac{2\times SENS\times SPEC}{SENS+SPEC}$$where TP is a true positive, TN is a true negative, FP is a false positive, and FN is a false negative. SENS stands for sensitivity, and SPEC stands for specificity.

Selected wavenumbers derived from SPA-QDA and GA-QDA for oesophageal stages [i.e., normal vs. inflammatory vs. Barrett’s vs. low-grade dysplasia (LGD) vs. high-grade dysplasia (HGD) vs. oesophageal adenocarcinoma (OAC)] were confirmed by a two-tailed non-parametric Mann–Whitney test (95% confidence interval). In addition, model robustness and uncertainty estimations were calculated for each developed model on the basis of misclassification probability estimations according to the bootstrapping method described in [25]. 

## 3. Results

Tissue specimens were categorised into the following groups for spectral analysis: *n* = 35 (normal), *n* = 13 (inflammatory), *n* = 26 (Barrett’s), *n* = 4 (LGD), *n* = 9 (HGD), and *n* = 22 (OAC). The average IR spectra for all oesophageal stages of disease appeared to overlap in the biochemical fingerprint region (1800 cm^−1^ to 900 cm^−1^) making it difficult to distinguish any subtle but significant differences (Figure 2A). Discriminant peaks were observed at around 1640 cm^−1^ (amide I absorption—predominantly the C=O stretching vibration of the amide C=O) [23], 1540 cm^−1^ (amide II of proteins) [28], and 1393 cm^−1^ (methylene deformation) [29]. Additional significant peaks were found at around 1690 cm^−1^ (peak of nucleic acids due to the base carbonyl stretching and ring breathing mode) [30], 1225 cm^−1^ (asymmetric phosphate stretching vibrations, *v*_as_PO_2_^−^) [30], and 1745 cm^−1^ (ester group (C=O) vibration of lipids) [28,31] (Appendix A). The spectral dataset was pre-processed using rubber-band baseline correction and normalisation to the amide I peak to further categorise and discriminate among all classifications to OAC (Figure 2B). The spectral noise observed in Figure 2A,B can be experimentally improved by increasing the number of scans, and the wavenumber differences can be enhanced by increasing the spectral resolution, which may lead to better elucidation. For the classification purpose, chemometric techniques (PCA-QDA, SPA-QDA, and GA-QDA) were adopted to classify normal vs. inflammatory vs. Barrett’s vs. LGD vs. HGD vs. OAC on the basis of their IR spectra.

The PCA-QDA model using the scores on six PCs (93.5% of the total data variance) demonstrated an average overall accuracy of 90.9% for all classes, an average sensitivity >71% for all classes (71.4% for normal and 100% for the other classes), an average specificity of >89% for all classes (89.5% for inflammatory and 100% for the other classes), and an average F-score >83% for all classes (83.3% for normal, 94.5% for inflammatory, and 100% for the other classes) (Table 1). This shows that some samples from the normal class were misclassified as inflammatory using PCA-QDA, but the remaining samples were all classified correctly (Table 2). The SPA-QDA model achieved 86.4% overall accuracy for all classes using six wavenumbers (Appendix A). The model misclassified all inflammatory samples as normal, but correctly classified all the other samples (sensitivity of 100% for all classes except inflammatory) (Table 2). 

The GA-QDA model based on 25 selected wavenumbers also misclassified most normal samples as inflammatory and all inflammatory samples as normal (Table 2). All the other classes were correctly classified with 100% sensitivity and specificity. This confusion between normal and inflammatory may be caused due to the similarity between these tissues. The best overall results were obtained for the PCA-QDA model, as shown in Table 1 and in the confusion table of Table 2.

In addition, model robustness was calculated on the basis of the misclassification probability for each model [25]. The average misclassification probability for each model was 0.322 (PCA-QDA), 0.311 (SPA-QDA), and 0.181 (GA-QDA). The misclassification probability ranges from 0 to 1, where values above 0.5 indicate higher probability of misclassification in future predictions (high model uncertainty and overfitting). The results obtained herein were, therefore, in a good range, indicating that the models are robust. Slightly lower uncertainty was observed for GA-QDA, confirming the tendency for overlapping between normal and inflammatory classes in future predictions due to their similarities, with good separation of the remaining groups of tissues. 

## 4. Discussion

Screening and surveillance in OAC and BO both require a skilled endoscopist for OGD for mucosal sampling and a pathologist for accurate histopathological examination. The development of a quick, convenient, and inexpensive method for detecting early malignancy in these specimens can guide further tissue biopsies and, thus, increase the yields of dysplasia detection. ATR-FTIR vibrational spectroscopy could be utilised as an adjunct to histology when the specific diagnosis of dysplasia is unclear/equivocal.

Fourier-transform infrared (FTIR) spectroscopy has been employed to study cancer in gastrointestinal tissues including stomach and intestinal tissue [26]. The technology is capable of differentiating between unaffected and malignant tissues by comparing spectra for changes in an array of diagnostic bands arising from phosphate, C–O, and CH stretching vibrational modes. Chemometric methods such as principal component analysis (PCA) and hierarchical clustering analysis (HCA) are commonly used to separate spectra of normal and neoplastic regions. Despite these technological improvements, biomedical applications of the FTIR-based analytical technique have not progressed. The majority of studies in this field have focused on ex vivo analysis.

The existing literature has explored ATR-FTIR spectroscopy and its use in the diagnosis and identification of oesophageal pre-malignant and malignant conditions. Wang et al. [27] identified significant protein, nucleic acid, sugar, and fat cell composition differences between normal and malignant oesophageal tissues, as a result of content changes in protein, nuclear acid, sugar, and fat composition in cells. Maziak et al. [9] built on work by Wang et al. and identified an increase in the nucleus-to-cytoplasm ratio and triglycerides and proteins from OAC specimens compared to those with squamous mucosa/normal oesophageal tissue specimens.

Quaroni and Casson [32] performed pilot work on BO tissue as a precursor to OAC using FTIR. The study published in 2009 found that intestinal metaplasia in BO tissues displayed characteristic IR absorption spectra of glycoproteins. Additionally, they established that these glycoprotein regions were more fragmented compared to OAC tissue specimens. 

Wang et al. [33] analysed premalignant (dysplastic) mucosa in BO using FTIR. The authors were able to classify normal squamous samples from ‘abnormal’ samples (any stage of Barrett’s) with a total accuracy of 92%, a sensitivity of 80%, and a specificity of 92% in 38 specimens. This also led to a better interobserver agreement between two gastrointestinal pathologists for dysplasia (κ = 0.72) vs. histology alone (κ = 0.52).

Old et al. [8] mapped 22 oesophageal tissue samples from 19 patients using FTIR. This group was able to classify between normal squamous samples and ‘abnormal’ samples (any stage of Barrett’s) with 100% sensitivity and specificity. The authors further concluded that FTIR could classify any grade of dysplastic Barrett’s (dysplasia or adenocarcinoma) with 95.6% sensitivity and 86.4% specificity. This highly accurate pathological classification can be achieved with FTIR measurement of frozen tissue sections in a clinically applicable timeframe.

This paper reports the use of ATR-FTIR coupled multivariate classification techniques (PCA-QDA, SPA-QDA, and GA-QDA) in identifying oesophageal stages of disease to adenocarcinoma has achieved excellent accuracy, sensitivity, and specificity, encouraging investigation of screening for other cancers with known markers in an ex vivo setting. Classification of normal squamous samples versus ‘abnormal’ samples (any stage of Barrett’s) was performed with 100% sensitivity and specificity using the classification models. These findings compare with the previous literature [8,33].

A significant finding from this study highlights the use of PCA-QDA, SPA-QDA, and GA-QDA in defining BO, LGD, HGD, and OAC with 100% sensitivity and specificity. This appears more accurate than previous studies [8,32,33].

A diagnosis of LGD is difficult to distinguish from inflammation histopathologically due to subtle cellular differences. The overall risk of progression of LGD to HGD is approximately 9% [11]. Given the large inter-observer variability in histopathological diagnosis in these groups, this highlights the need for a distinct method of categorisation. As a clear diagnosis of LGD is the only current reliable marker which can sub-select those at higher risk of progression to malignancy [7], using ATR-FTIR could be a crucial diagnostic adjunct in a histopathologist’s armoury.

ATR-FTIR spectroscopy has been utilised to identify neoplasia in ovarian [34], skin [35], and pancreatic tissue [36]. Over the past 20 years, there has been a large shift towards biofluid sampling to identify biomarkers of malignancies. Paraskevaidi et al. [37] used ATR-FTIR to analyse urine samples from women with endometrial and ovarian cancer, as well as from healthy individuals. The authors found high levels of accuracy for both endometrial (95% sensitivity, 100% specificity) and ovarian cancer (100% sensitivity, 96.3% specificity). ATR-FTIR spectroscopy has also been used in conjunction with specific serum assays to stratify brain tumour histological subtypes [38]. In the field of oesophageal disease, Maitra et al. [22] utilised ATR-FTIR spectroscopy to predict six oesophageal stages in four different biofluids (plasma, saliva, serum, and urine). PCA-QDA and GA-QDA models were found to give the best class differentiation compared to the SPA-QDA. The GA-QDA model utilised in plasma samples was able to predict all stages of disease to OAC with 100% accuracy, sensitivity, and specificity. For this model, several selected wavenumbers appeared to be of particular interest, especially at 999 cm^−1^ and 1381 cm^−1^, representing the ring stretching vibrations mixed strongly with CH in-plane bending and C–O stretching, respectively.

Biofluid analysis has been a favourable technique in vibrational spectroscopy for its high throughput. Measurement conditions can be controlled, and the required sample volume is small. Major limitations are of measurement variations due to post-collection procedures such as sample dilution, storage at cold temperatures, drying effects, and the addition of anticoagulants [39].

The morphological classification of certain tumours is still challenging even with the advent of staining and other histopathological adjuncts. The recent literature suggests that vibrational spectroscopy has molecular sensitivity towards biochemical changes in tissue rivalling immunohistochemistry [40]. Vibrational spectroscopy detects changes in the metabolome and proteome. The techniques of ATR-FTIR and Raman spectroscopy are quick and cheap, and they require little tissue material.

The literature highlights only a few studies on identifying benign disorders of the oesophagus using ATR-FTIR spectroscopy. We propose that an earlier diagnosis of dysplasia in BO patients using ATR-FTIR in tissue as an adjunct would lead to less invasive intervention and may subsequently reduce the number of endoscopies patients undergo in the future, thus being cost-effective.

## 5. Conclusions

The use of ATR-FTIR spectroscopy of oesophageal tissue coupled with multivariate classification algorithms (PCA-QDA, SPA-QDA, and GA-QDA) results in a powerful alternative approach to accurate histopathological diagnosis. Classification of normal squamous samples versus ‘abnormal’ samples (any stage of Barrett’s) was performed with 100% sensitivity and specificity using the classification models. These findings complement findings from previous literature. 

This study was performed using standard operating protocols with tissue sample preparation, as well as ATR-FTIR spectroscopy measurement [16]. Furthermore, standardised chemometric evaluation and analysis techniques for predictive categorisation were utilised [17]. The major limitation of this thesis is the relatively small number of tissues analysed. This would need further multicentre, multi-laboratory analysis for validation, repeatability, and reproducibility of spectral datasets. 

The technology has been Importantly accurate in diagnosing BO, LGD, HGD, and OAC with high values of accuracy, sensitivity, and specificity (100%). Making an accurate diagnosis of dysplasia earlier in the oncogenic process would mean less invasive treatments and reduce the burden of morbidity and mortality associated with OAC.

Standard operating protocols have been utilised in the pre-processing stage of biofluid analysis with vibrational spectroscopy. Non-modifiable biological factors and patient lifestyle factors would need to be considered when performing future spectroscopic analysis. It is unclear as to how individual factors such as a patient’s diet, comorbidities, and lifestyle factors would have a pre-analytical influence on laboratory parameters and subsequently affect IR spectral data in human tissue. This would warrant substantial further investigation.

The cost-effectiveness of surveillance in Barrett’s oesophagus is often questioned because the rate of conversion from Barrett’s oesophagus to adenocarcinoma is only 0.5% per year [14]. Nevertheless, arguments for frequent surveillance include that most adenocarcinomas result within Barrett’s oesophagus segments, and the risk of OAC is about 30–40 times higher in patients with Barrett’s oesophagus compared to those without [14].

Vibrational spectroscopy identifies molecular data that can be rapidly acquired without the need for specialised sample preparation. This could potentially streamline analyses for interventions in many fields in the modern-day National Health Service. 

An earlier diagnosis of dysplasia in Barrett’s oesophagus patients would ultimately enable fewer invasive and more expensive surgical options. Optical techniques combined with vibrational spectroscopy could not only aid in differentiating grades of dysplasia in tissue but also identify potential future biomarkers.

Translational studies moving vibrational spectroscopy into the clinical field have been preliminary. Relying on vibrational spectroscopy as a definite diagnostic or prognostic tool requiring further directive imaging is up for deliberation. Limitations of using vibrational spectroscopy include the need for sensitive and highly optimised instrumentation, as well as theoretical heating effects with analysis. A clinician needs to weigh the advantages with the disadvantages of using this technology prior to advising further direct invasive or non-invasive testing.

## Figures and Tables

**Figure 1 jpm-13-01277-f001:**
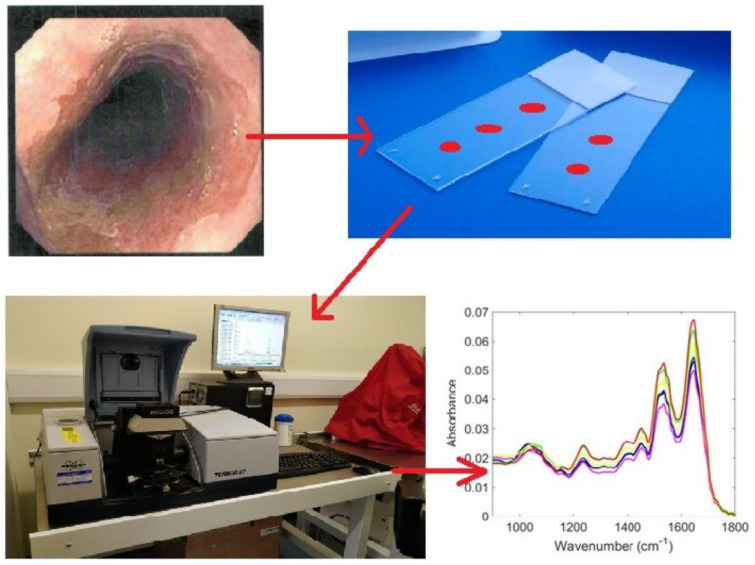
Graphical abstract demonstrating how oesophageal tissue is processed through ATR-FTIR spectroscopy in order to detect oesophageal transformation stages to adenocarcinoma.

**Figure 2 jpm-13-01277-f002:**
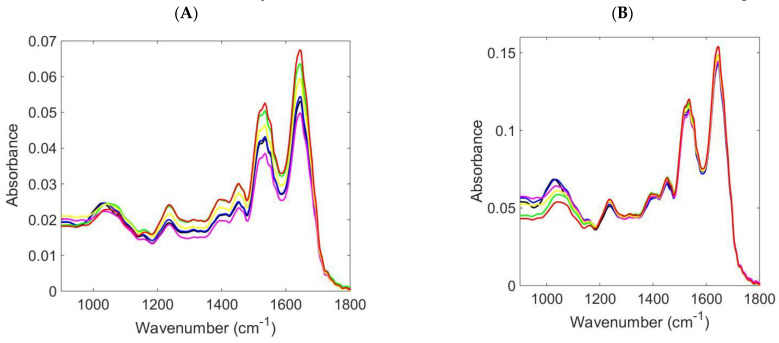
Comparison of normal/inflammatory/Barrett’s oesophagus/LGD/HGD/OAC oesophageal stages using tissue samples for ATR spectroscopy: (**A**) average raw spectra in the ATR region of 1800 cm^−1^ to 900 cm^−1^, and (**B**) average pre-processed ATR spectra obtained from all stages segregated into normal (black colour) vs. inflammatory (blue colour) vs. Barrett’s oesophagus (green colour) vs. LGD (yellow colour) vs. HGD (magenta colour) vs. OAC (red colour).

**Table 1 jpm-13-01277-t001:** Figures of merit (FOM) showing the average accuracy, sensitivity, specificity, and F-scores for the multivariate classification methods (PCA-QDA, SPA-QDA, and GA-QDA) to distinguish among normal vs. inflammatory vs. Barrett’s oesophagus vs. LGD vs. HGD vs. OAC tissue samples. Results were calculated on the basis of a Monte Calo cross-validation with 1000 iterations, leaving 20% of samples out for validation.

FOM	PCA-QDA (Accuracy = 90.9%)					
	Normal	Inflammatory	Barrett’s Oesophagus	LGD	HGD	OAC
Sensitivity (%)	71.4	100.0	100.0	100.0	100.0	100.0
Specificity (%)	100.0	89.5	100.0	100.0	100.0	100.0
F-Score (%)	83.3	94.5	100.0	100.0	100.0	100.0
**FOM**	**SPA-QDA (Accuracy = 86.4%)**					
	**Normal**	**Inflammatory**	**Barrett’s** **oesophagus**	**LGD**	**HGD**	**OAC**
Sensitivity (%)	100.0	0.00	100.0	100.0	100.0	100.0
Specificity (%)	80.0	100.0	100.0	100.0	100.0	100.0
F-Score (%)	88.9	0.00	100.0	100.0	100.0	100.0
**FOM**	**GA-QDA (Accuracy = 63.6%)**					
	**Normal**	**Inflammatory**	**Barrett’s** **oesophagus**	**LGD**	**HGD**	**OAC**
Sensitivity (%)	28.6	0.00	100.0	100.0	100.0	100.0
Specificity (%)	80.0	73.7	100.0	100.0	100.0	100.0
F-Score (%)	42.1	0.00	100.0	100.0	100.0	100.0

**Table 2 jpm-13-01277-t002:** Confusion matrices showing the average validation output for the multivariate classification methods (PCA-QDA, SPA-QDA, and GA-QDA) to distinguish among normal vs. inflammatory vs. Barrett’s oesophagus vs. LGD vs. HGD vs. OAC tissue samples. Results were calculated on the basis of a Monte Calo cross-validation with 1000 iterations, leaving 20% of samples out for validation.

Real/Predicted	PCA-QDA					
	Normal	Inflammatory	Barrett’s Oesophagus	LGD	HGD	OAC
**Normal (*n* = 35)**	25	10	0	0	0	0
**Inflammatory (*n* = 13)**	0	13	0	0	0	0
**Barrett’s oesophagus** ** (*n* = 26)**	0	0	26	0	0	0
**LGD (*n* = 4)**	0	0	0	4	0	0
**HGD (*n* = 9)**	0	0	0	0	9	0
**OAC (*n* = 22)**	0	0	0	0	0	22
**Real/Predicted**	**SPA-QDA**					
	**Normal**	**Inflammatory**	**Barrett’s** **oesophagus**	**LGD**	**HGD**	**OAC**
**Normal (*n* = 35)**	35	0	0	0	0	0
**Inflammatory (*n* = 13)**	13	0	0	0	0	0
**Barrett’s oesophagus** **(*n* = 26)**	0	0	26	0	0	0
**LGD (*n* = 4)**	0	0	0	4	0	0
**HGD (*n* = 9)**	0	0	0	0	9	0
**OAC (*n* = 22)**	0	0	0	0	0	22
**Real/Predicted**	**GA-QDA**					
	**Normal**	**Inflammatory**	**Barrett’s** **oesophagus**	**LGD**	**HGD**	**OAC**
**Normal (*n* = 35)**	10	25	0	0	0	0
**Inflammatory (*n* = 13)**	13	0	0	0	0	0
**Barrett’s oesophagus** **(*n* = 26)**	0	0	26	0	0	0
**LGD (*n* = 4)**	0	0	0	4	0	0
**HGD (*n* = 9)**	0	0	0	0	9	0
**OAC (*n* = 22)**	0	0	0	0	0	22

## Data Availability

Anonymised data and spectral datasets are available at the School of Pharmacy and Biomedical Sciences Laboratories at the University of Central Lancashire, UK campus.

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
