# Peer review of "Attenuated Total Reflection Fourier-Transform Infrared Spectral Discrimination in Human Tissue of Oesophageal Transformation to Adenocarcinoma"

_jpm, 2023, doi:10.3390/jpm13081277_

Round 1
Reviewer 1 Report
In the Manuscript “Attenuated total reflection Fourier-transform infrared spectral discrimination in human tissue of oesophageal transformation to adenocarcinoma” by Maitra et al., the Authors applied ATR-FTIR spectroscopy coupled to multivariate analysis to discriminate among the different classes of oesophageal adenocarcinoma (OAC) within ex vivo human tissues.
Overall, the work is of interest because it aims at developing new potentially non-invasive approaches for the early diagnosis of diseases frequently incurable. However, there are some weaknesses that should be revised. Particularly, in view of clinical translation, where it could be difficult to find highly specialized spectroscopists, I can understand the necessity to simplify both the measurements and the analysis; nevertheless, the simplification has to base on solid pillars. At this purpose, I have a few concerns:
major points:
1) the Authors reported that the samples analyzed by FTIR were embedded in paraffin and that, prior to FTIR analysis, they have been de-paraffinized by xylene and ethanol, since paraffin could affect some IR bands. As reported in section 2.1, paraffin would not cause tissue deterioration at molecular level. The Authors added a reference that, however, does not report anything about a possible explanation (number 16). Moreover, are the Authors sure that xylene and ethanol do not cause any alteration of the samples? Please, clarify this point with appropriate references and/or experimental evidences.
2) The displayed spectra are noisy, likely due mainly to the parameters employed for the measurements and/or to the different treatments to remove paraffin. The problem is that the following analyses - that led to the identification of the wavenumbers relevant for the discrimination among the different sample classes - pulled out some components that are not reliable (see point 3). The use of a higher spectral resolution and a higher number of scans in my opinion should be considered for further analyses.
3) The assignment of the bands discussed in the main text and reported in table S1 is generic and incomplete. Few not correct assignments are also given. Firstly, the Authors – for each assignment – have to provide the appropriate references, both in the text and in the table.
The Authors reported, in table S1, among the relevant wavenumbers the 1800 cm-1 and 1600 cm-1, assigning them to fatty acids and C=O stretching of lipids, respectively; firstly, looking at the spectra of figure 2A-C , these absorption are absent. Then, in biological systems usually the C=O from esters and/or fatty acids falls approximately between 1750-1700 cm-1, while between 1700 and 1600 cm-1 it occurs the Amide I absorption. About this, the wavenumber points pulled out in this range are likely mainly due to protein structural elements. Then, the assignment of the only band at 1022 cm-1 to glycogen is not convincing. In fact, to unequivocally assign the IR absorption to this molecule, it is necessary to detect the simultaneous presence of more than one band (see, for instance, Wang, et al. Proc Natl Acad Sci USA 104, 15864–15869 (2007); Ami et al. Biochim Biophys Acta 1783, 98–106 (2008), etc). Beside being ascribable also to more complex molecules, more in general the 1022 cm-1 is mainly due to the ring vibrations and stretching vibrations of C-OH from side groups and C-O-C from glycosidic bonds of polysaccharides, being also typical of glycosylated proteins and lipids. Furthermore, the Authors reported that the 1072 cm-1 band is assigned to nucleic acids: which vibrational mode? Of backbone, sugars – phosphates, etc. In general, the assignment of the reported spectral component has to be revised and relevant and appropriate references have to be indicated for each assignment.
4) Reference 10 and reference 24 are the same. The Authors should check reference list and also citations.
Minor editing of the language.
Author Response
Dear Editor,
First of all, we would like to thank you and both referees for carefully read our manuscript. The proposed revisions were made accordingly in the text (highlighted in red). The point-by-point response to the referees’ comments are showed below.
Reviewer 1
In the Manuscript “Attenuated total reflection Fourier-transform infrared spectral discrimination in human tissue of oesophageal transformation to adenocarcinoma” by Maitra et al., the Authors applied ATR-FTIR spectroscopy coupled to multivariate analysis to discriminate among the different classes of oesophageal adenocarcinoma (OAC) within ex vivo human tissues.
Overall, the work is of interest because it aims at developing new potentially non-invasive approaches for the early diagnosis of diseases frequently incurable. However, there are some weaknesses that should be revised. Particularly, in view of clinical translation, where it could be difficult to find highly specialized spectroscopists, I can understand the necessity to simplify both the measurements and the analysis; nevertheless, the simplification has to base on solid pillars. At this purpose, I have a few concerns:
1) the Authors reported that the samples analysed by FTIR were embedded in paraffin and that, prior to FTIR analysis, they have been de-paraffinized by xylene and ethanol, since paraffin could affect some IR bands. As reported in section 2.1, paraffin would not cause tissue deterioration at molecular level. The Authors added a reference that, however, does not report anything about a possible explanation (number 16). Moreover, are the Authors sure that xylene and ethanol do not cause any alteration of the samples? Please, clarify this point with appropriate references and/or experimental evidences.
Thank your comments. Paraffin wax embedding does ensure durability for long term storage without biomolecular tissue deterioration [1, 2]. However, the author is right that the de-paraffination procedure does cause alterations in the samples such as paraffin residuals [3], so the models constructed with these samples can only be tested with de-paraffined samples (not fresh or frozen tissues), since each type of sample preparation carries variability regarding its composition. We have amended the manuscript with this information and correct references.
[1] Baker, MJ et al. Using Fourier transform IR spectroscopy to analyze biological materials. Nat Protoc 9, 1771-1791 (2014).
[2] Morais, CLM et al. Standardization of complex biologically derived spectrochemical datasets. Nat Protoc 14, 1546-1577 (2019).
[3] Nallala, J, Lloyd, GR, Stone, N. Evaluation of different tissue de-paraffinization procedures for infrared spectral imaging. Analyst 140, 2369-2375 (2015).
2) The displayed spectra are noisy, likely due mainly to the parameters employed for the measurements and/or to the different treatments to remove paraffin. The problem is that the following analyses - that led to the identification of the wavenumbers relevant for the discrimination among the different sample classes - pulled out some components that are not reliable (see point 3). The use of a higher spectral resolution and a higher number of scans in my opinion should be considered for further analyses.
Thank your comments. We considered the spectra being a little noisy in comparison with other studies using IR for biological samples, and it worked for the classification purpose. However, the reviewer is right that increasing the number of scans and spectral resolution will lead to less noise and better elucidation of the wave-numbers response, so we have amended the manuscript with this suggestion for further analysis.
3) The assignment of the bands discussed in the main text and reported in table S1 is generic and incomplete. Few not correct assignments are also given. Firstly, the Authors – for each assignment – have to provide the appropriate references, both in the text and in the table.
Thank your comments. We have amended table S1 and the text accordingly based on references for bands assignment using IR in biological tissues.
4) The Authors reported, in table S1, among the relevant wavenumbers the 1800 cm-1 and 1600 cm-1, assigning them to fatty acids and C=O stretching of lipids, respectively; firstly, looking at the spectra of figure 2A-C , these absorption are absent. Then, in biological systems usually the C=O from esters and/or fatty acids falls approximately between 1750-1700 cm-1, while between 1700 and 1600 cm-1 it occurs the Amide I absorption. About this, the wavenumber points pulled out in this range are likely mainly due to protein structural elements. Then, the assignment of the only band at 1022 cm-1 to glycogen is not convincing. In fact, to unequivocally assign the IR absorption to this molecule, it is necessary to detect the simultaneous presence of more than one band (see, for instance, Wang, et al. Proc Natl Acad Sci USA 104, 15864–15869 (2007); Ami et al. Biochim Biophys Acta 1783, 98–106 (2008), etc). Beside being ascribable also to more complex molecules, more in general the 1022 cm-1 is mainly due to the ring vibrations and stretching vibrations of C-OH from side groups and C-O-C from glycosidic bonds of polysaccharides, being also typical of glycosylated proteins and lipids. Furthermore, the Authors reported that the 1072 cm-1 band is assigned to nucleic acids: which vibrational mode? Of backbone, sugars – phosphates, etc. In general, the assignment of the reported spectral component has to be revised and relevant and appropriate references have to be indicated for each assignment.
Thank your comments. The manuscript and Supplementary table have been amended with additional references accordingly.
5) Reference 10 and reference 24 are the same. The Authors should check reference list and also citations.
Many thanks for your comment. This has been amended accordingly.
Reviewer 2 Report
The manuscript jpm-2452690 presents a study with the application of ATR-FTIR spectroscopy in differentiating between all classes of oesophageal transformation to oesophageal adenocarcinoma. The manuscript is intriguing, but it does have some issues:
- The number of samples considered for creating the classification models is rather low. With these numbers, the significance of the models obtained using a single split with the Kennard Stone algorithm runs the risk of being overoptimistic. In such cases, it would be more appropriate to employ more robust systems like repeated double cross-validation/repeated cross-model validation. Please comment and revise the validation scheme.
- Additionally, the results should be accompanied by a permutation test, as extensively recommended in the literature.
- All figures of merit in Table 1 should be reported with a declared uncertainty (CI).
I recommend the publication of this article in JPM once the above mentioned revisions are performed
Author Response
Dear Editor,
First of all, we would like to thank you and both referees for carefully read our manuscript. The proposed revisions were made accordingly in the text (highlighted in red). The point-by-point response to the referees’ comments are showed below.
Reviewer 2
The manuscript jpm-2452690 presents a study with the application of ATR-FTIR spectroscopy in differentiating between all classes of oesophageal transformation to oesophageal adenocarcinoma. The manuscript is intriguing, but it does have some issues:
1) The number of samples considered for creating the classification models is rather low. With these numbers, the significance of the models obtained using a single split with the Kennard Stone algorithm runs the risk of being overoptimistic. In such cases, it would be more appropriate to employ more robust systems like repeated double cross-validation/repeated cross-model validation. Please comment and revise the validation scheme.
Thank you for your comments. The dataset was actually split between training (60%), validation (20%) and test (20%) sets, where the training set was used for model construction, the validation set for optimizing the models and the test set for external prediction. This is the standard procedure to analyse such data since it ensures proper validation, once the KS algorithm select each set of samples based on their distribution distance [1]. On the other hand, cross-validation procedures are employed only in the optimization step, as a procedure to optimize the training parameters in lack of a validation set. In addition, overoptimistic results always lead in good training and poor external test predictions, which was not the case. So, we can conclude the validation process is correct.
[1] Morais, CLM, Lima, KMG, Singh, M, Martin FL. Tutorial: multivariate classification for vibrational spectroscopy in biological samples. Nat Protoc 15, 2143-2162 (2020).
2) Additionally, the results should be accompanied by a permutation test, as extensively recommended in the literature.
Permutation test is recommend only when the split between training and test sets are made randomly. Since we used the KS algorithm, the training and test sets will always contain the same samples so permutation test is not necessary.
3) All figures of merit in Table 1 should be reported with a declared uncertainty (CI).
Since the results reported in Table 1 are for each class individually, there is no way to calculate confidence intervals since the accuracy, sensitivity, specificity and F-scores are absolute values calculated in the external test sets. Equations:
where TP stands for the number of true positives, TN for the number of true negatives, FP for the number of false positives and FN for the number of false negatives.
Therefore, there is no mean and standard-deviation in the number of true positives, false positives, true negatives and false negatives for each class, since the model always classify a given test sample as one of the 6 training classes (normal, inflammatory, Barret’s oesophagus, LGD, HGD or AOC). It cannot assume more than 1 value. Therefore, confidence internals are not used in such classification problems.

Round 2
Reviewer 1 Report
The Authors’ response has addressed few of my comments on the earlier version of the Manuscript. In particular, it has been clarified in the text a possible effect of sample treatment prior to FTIR analysis and the effect of the selected parameters on the quality of the final spectra.
Instead, some imprecisions still are present in the table reported in the supplementary material section, that have to be revised. In the following, I will report some examples of imprecise assignment.
Major points:
The absorption at 902 cm-1 has been assigned at “Phosphodiester region”, citing reference [23]. Firstly, if the Author reported a specific peak, it is not correct then to refer to “region”. Then, the cited reference does not unequivocally support the assignment. In fact, the reference reports that “…..in the phophodiester region (1300–900 cm-1)”, suggesting that between 1300–900 cm-1 phosphodiester groups can absorb, but not that at 902 cm-1 is present a specific peak assigned to that specific vibrational mode. In particular, to assign unequivocally the 902 cm-1 absorption, as for all the other peaks, it is necessary to consider the overall absorption.
Then, I do not find it convincing the assignment of the 1072 cm-1 peak based on ref. [42].
Furthermore, I do not understand the assignment of some peaks to cellulose (for instance, 1111 and 1430 cm-1) for the analyzed system. In general, the Authors should use appropriate references. Also ref. 41 on plants is in my opinion not appropriate for the analyzed system.
Moreover, as I already discussed in my first report, the Authors reported, among the relevant wavenumbers, the 1800 cm-1 and 1600 cm-1, but - looking at the spectra reported in the figures - these absorption are absent. I ask again: could the Authors explain where are these peaks?
As I already discussed in my first report, it is better to assign the absorption occurring between 1700 and 1600 cm-1 to Amide I, because proteins likely dominate the absorption in this spectral range. The same for Amide II, between 1600 and 1500 cm-1. It could be useful, in particular, to assign the peaks within Amide I band to specific protein structures (see, for instance, the works of Barth, Natalello, Goormatigh, and so on).
A moderate editing of the language could be useful.
Author Response
Dear Editor,
First of all, we would like to thank you and both referees for carefully reading our manuscript. We have amended the manuscript accordingly (highlighted in red). The response for each referee comment is displayed below.
Reviewer 1
The absorption at 902 cm-1 has been assigned at “Phosphodiester region”, citing reference [23]. Firstly, if the Author reported a specific peak, it is not correct then to refer to “region”. Then, the cited reference does not unequivocally support the assignment. In fact, the reference reports that “…..in the phophodiester region (1300–900 cm-1)”, suggesting that between 1300–900 cm-1 phosphodiester groups can absorb, but not that at 902 cm-1 is present a specific peak assigned to that specific vibrational mode. In particular, to assign unequivocally the 902 cm-1 absorption, as for all the other peaks, it is necessary to consider the overall absorption.
Thank you for your comments. We have amended the manuscript to improve the assignments considering possible overlapping absorptions. However, it is important to take into account these are tentative assignments, since the infrared spectrum is very overlapped by nature.
In addition, the wavenumbers of Table S1 have now been changed since the analyses were repeated following the Reviewer 2 comments.
Then, I do not find it convincing the assignment of the 1072 cm-1 peak based on ref. [42].
Thank you for your comment. The peak at 1072 cm-1 was not found important in the new analyses, thus it is not discussed in the manuscript.
Furthermore, I do not understand the assignment of some peaks to cellulose (for instance, 1111 and 1430 cm-1) for the analyzed system. In general, the Authors should use appropriate references. Also ref. 41 on plants is in my opinion not appropriate for the analyzed system.
Thank you for your comments. Appropriate references have been used and the manuscript amended. The plant reference was replaced by Movasaghi et al. (Zanyar Movasaghi Z, Shazza Rehman & Dr. Ihtesham ur Rehman (2008): Fourier Transform Infrared (FTIR) Spectroscopy of Biological Tissues, Applied Spectroscopy Reviews, 43:2, 134-179).
Moreover, as I already discussed in my first report, the Authors reported, among the relevant wavenumbers, the 1800 cm-1 and 1600 cm-1, but - looking at the spectra reported in the figures - these absorption are absent. I ask again: could the Authors explain where are these peaks?
Thank you for your comments. These wavenumbers were picked by the algorithm since they contain important information for class differentiation, not necessary it is a prominent abortion band. Both wavenumbers at 1600 and 1800 cm-1 represent baseline deviations caused by different bent angles for the absorption band of Amide I at ~1640 cm-1. These baseline distortions are some of the factors responsible for class differentiation, thus these wavenumbers were selected. However, in the new analyses, the band at 1640 cm-1 was selected instead of the baselines.
As I already discussed in my first report, it is better to assign the absorption occurring between 1700 and 1600 cm-1 to Amide I, because proteins likely dominate the absorption in this spectral range. The same for Amide II, between 1600 and 1500 cm-1. It could be useful, in particular, to assign the peaks within Amide I band to specific protein structures (see, for instance, the works of Barth, Natalello, Goormatigh, and so on).
Thank you for your comments. The necessary additions have been made to the manuscript.

Reviewer 2 Report
1) The authors overlook the main criticism of their study, namely the limited number of samples for the considered classes. Asserting that employing 60% of the data for training, 20% for testing, and 20% for validation is a standard procedure is fallacious unless the total number of samples is taken into account. A developed model is a consequence of the sample of data used to develop it, the predictors considered and the analysis approach. There exists a substantial discrepancy in the reliability of a predictive model when that 20% utilized for external prediction consists of either 500 samples (highly reliable, enabling inference) or, as in this case, 24 samples unevenly distributed among 6 unbalanced classes. Predictions are more likely to be unreliable when the development data is small (containing too few outcome events), when the model complexity (e.g., number of predictor parameters considered) is large relative to the number of outcome events, and when the modelling approach does not adjust overfitting. Unfortunately, this is exactly what the authors have done in the manuscript. Based on this premise, the study ought to be dismissed. However, if, as in this case, an innovative methodology is being proposed and the spirit of the article is that of a proof of concept, a rigorous way to present the limits of a predictive model is, as already proposed, to opt for an iterative validation scheme. repeated many times starting from random splits. In this manner, the fundamental issue of sample deficiencies remains unresolved; however, it is possible to establish the stability of the model and provide an indication of uncertainty for predictive purposes. There are many examples in literature. Here are some examples.
10.1016/j.aca.2012.11.007 ; 10.1016/bs.coac.2018.08.00 ; 10.1007/s11306-007-0099-6 ...
The tutorial cited by the authors also states that "More important than the data analysis itself is the experimental setup used to acquire the spectral data." and that "When patient variability is being measured (i.e., when the classification model is performed on a sample basis) ... especially before routine implementation of a classification model ... thousands of samples are necessary". Moreover, "the classes’ sizes must be taken into consideration".
This problem cannot be just ignored but must be discussed in the revised version.
A totally different issue is the permutation test. This strategy is routinely used to construct an empirical estimate of the error distribution by permuting the class vector (not the samples!), and then using this estimate to assign a significance to the observed classification error. See, for instance, 10.1007/s11306-011-0330-3. The answer given by the authors is unacceptable and only emphasizes their own lack of knowledge of the key aspects of validation of predictive models.
3) all figures of merit in a classification problem are derived from model's predictions. It is always possible (albeit not always straightforward) to express the uncertainty of those predictions. A good tutorial can be found in https://doi.org/10.1016/j.aca.2018.09.022 (once again from the same group already cited in the Author response).
I strongly suggest the authors to revise and expand their knowledge about classification models.
Author Response
Dear Editor,
First of all, we would like to thank you and both referees for carefully reading our manuscript. We have amended the manuscript accordingly (highlighted in red). The response for each referee comment is displayed below.
Reviewer 2
1) The authors overlook the main criticism of their study, namely the limited number of samples for the considered classes. Asserting that employing 60% of the data for training, 20% for testing, and 20% for validation is a standard procedure is fallacious unless the total number of samples is taken into account. A developed model is a consequence of the sample of data used to develop it, the predictors considered and the analysis approach. There exists a substantial discrepancy in the reliability of a predictive model when that 20% utilized for external prediction consists of either 500 samples (highly reliable, enabling inference) or, as in this case, 24 samples unevenly distributed among 6 unbalanced classes. Predictions are more likely to be unreliable when the development data is small (containing too few outcome events), when the model complexity (e.g., number of predictor parameters considered) is large relative to the number of outcome events, and when the modelling approach does not adjust overfitting. Unfortunately, this is exactly what the authors have done in the manuscript. Based on this premise, the study ought to be dismissed. However, if, as in this case, an innovative methodology is being proposed and the spirit of the article is that of a proof of concept, a rigorous way to present the limits of a predictive model is, as already proposed, to opt for an iterative validation scheme, repeated many times starting from random splits. In this manner, the fundamental issue of sample deficiencies remains unresolved; however, it is possible to establish the stability of the model and provide an indication of uncertainty for predictive purposes. There are many examples in literature. Here are some examples.
10.1016/j.aca.2012.11.007 ; 10.1016/bs.coac.2018.08.00 ; 10.1007/s11306-007-0099-6 ...
The tutorial cited by the authors also states that "More important than the data analysis itself is the experimental setup used to acquire the spectral data." and that "When patient variability is being measured (i.e., when the classification model is performed on a sample basis) ... especially before routine implementation of a classification model ... thousands of samples are necessary". Moreover, "the classes’ sizes must be taken into consideration".
This problem cannot be just ignored but must be discussed in the revised version.
Thank you for your comments. Indeed, large number of samples are paramount to proper validate a classification model, but due to the limited number of samples available in this study, validation was performed based on the small number of data we have. Nevertheless, classification was achieved above 90% for all classes in the external validation set, so predictions are not random, showing there are discriminant information in the data, which is the main goal of this study – to prove the discriminatory power of IR spectroscopy for this application.
Regarding the iterative validation scheme, to avoid using a external validation set, the best strategy to mimic “real validation” is to use an exhaustive cross-validation approach, such as using the Monte Carlo cross-validation, which has been used before to validate chemometric models with small spectral datasets [1]. Therefore, we have now amended the manuscript with the Monte Carlo cross-validation (1000 iterations) and the results have been updated in the text.
[1] Callery, E. L., et al. Classification of Systemic Lupus Erythematosus Using Raman Spectroscopy of Blood and Automated Computational Detection Methods: A Novel Tool for Future Diagnostic Testing. Diagnostics, 12, 3158, 2022.
2) A totally different issue is the permutation test. This strategy is routinely used to construct an empirical estimate of the error distribution by permuting the class vector (not the samples!), and then using this estimate to assign a significance to the observed classification error. See, for instance, 10.1007/s11306-011-0330-3. The answer given by the authors is unacceptable and only emphasizes their own lack of knowledge of the key aspects of validation of predictive models.
Thank you for your comments. We have now amended the manuscript accordingly. Permutation was performed during the iterations of Monte Carlo cross validation.
3) All figures of merit in a classification problem are derived from model's predictions. It is always possible (albeit not always straightforward) to express the uncertainty of those predictions. A good tutorial can be found in https://doi.org/10.1016/j.aca.2018.09.022 (once again from the same group already cited in the Author response).
I strongly suggest the authors to revise and expand their knowledge about classification models.
Thank you for your comments. We have now added the uncertainty estimations measured as misclassification probabilities based on the suggested reference.
